# Activation of P2X7 Receptor Mediates the Abnormal Ovulation Induced by Chronic Restraint Stress and Chronic Cold Stress

**DOI:** 10.3390/biology13080620

**Published:** 2024-08-15

**Authors:** Xiang Fan, Jing Wang, Yinyin Ma, Dandan Chai, Suo Han, Chuyu Xiao, Yingtong Huang, Xiaojie Wang, Jianming Wang, Shimeng Wang, Li Xiao, Chunping Zhang

**Affiliations:** 1Department of Cell Biology, School of Basic Medical Sciences, Jiangxi Medical College, Nanchang University, Nanchang 330031, China; 18309471993@163.com (X.F.); 17339366776@163.com (Y.M.); hansuo2023@163.com (S.H.); 15170234392@163.com (C.X.); huangyingtong21@163.com (Y.H.); 15137921368@163.com (X.W.); wangjianming178@163.com (J.W.); wangshim18@163.com (S.W.); xiaoli@ncu.edu.cn (L.X.); 2Institute of Rehabilitation Science, Shaanxi Provincial Rehabilitation Hospital, Xi’an 710065, China; 3Department of Microbiology, School of Basic Medical Sciences, Jiangxi Medical College, Nanchang University, Nanchang 330031, China; wangj8001@163.com; 4Shangrao People’s Hospital, Shangrao 334000, China; chaidandan1129@163.com

**Keywords:** chronic restraint stress, chronic cold stress, corpus luteum, ovulation, P2X7R, fibrosis

## Abstract

**Simple Summary:**

Chronic stress significantly impacts physical and mental health and has also been shown to affect female reproduction, but the underlying mechanisms are not well understood. Our research focused on the P2X7 receptor, a protein involved in the pathological response under long-term stress. Using mouse models subjected to chronic restraint and cold stress, we observed reduced ovulation rates, a key indicator of reproductive health. We discovered that stress increases the expression of the P2X7 receptor in the ovaries. Treatment with P2X7R inhibitor partially rescued the ovulation rate of the two chronic stress models. This study reveals that managing stress and potentially targeting the P2X7 receptor could help improve reproductive health in women experiencing chronic stress, offering a new approach to treating stress-related reproductive issues.

**Abstract:**

Chronic stress has become a major problem that endangers people’s physical and mental health. Studies have shown that chronic stress impairs female reproduction. However, the related mechanism is not fully understood. P2X7 receptor (P2X7R) is involved in a variety of pathological changes induced by chronic stress. Whether P2X7R is involved in the effect of chronic stress on female reproduction has not been studied. In this study, we established a chronic restraint stress mouse model and chronic cold stress mouse model. We found that the number of corpora lutea was significantly reduced in the two chronic stress models. The number of corpora lutea indirectly reflects the ovulation, suggesting that chronic stress influences ovulation. P2X7R expression was significantly increased in ovaries of the two chronic stress models. A superovulation experiment showed that P2X7R inhibitor A-438079 HCL partially rescued the ovulation rate of the two chronic stress models. Further studies showed that activation of P2X7R signaling inhibited the cumulus expansion and promoted the expression of NPPC in granulosa cells, one key negative factor of cumulus expansion. Moreover, sirius red staining showed that the ovarian fibrosis was increased in the two chronic stress models. For the fibrosis-related factors, TGF-β1 was increased and MMP2 was decreased. In vitro studies also showed that activation of P2X7R signaling upregulated the expression of TGF-β1 and downregulated the expression of MMP2 in granulosa cells. In conclusion, P2X7R expression was increased in the ovaries of the chronic restraint-stress and chronic cold-stress mouse models. Activation of P2X7R signaling promoted NPPC expression and cumulus expansion disorder, which contributed to the abnormal ovulation of the chronic stress model. Activation of P2X7R signaling is also associated with the ovarian fibrosis changes in the chronic stress model.

## 1. Introduction

Stress refers to an intrinsic or extrinsic stimulus, which can induce biological response. Depending on the type, timing and severity of the imposed stimulus, the effects of stress on the human body vary a lot, ranging from minor alterations in homeostasis to severe injuries and even death [1]. Based on the duration of stress, it is divided into acute and chronic stress. Typically, acute stress imposed on young and healthy individuals may lead to adaption and does not cause serious harm [2]. However, clinical studies and animal studies showed that long-term chronic stress is a crucial risk factor for numerous types of disorders, including cardiovascular disorders, neurodegenerative disorders, depression, endocrinal disorders, cancers, and so on [3,4,5,6,7]. In fact, compared with males, females are more susceptible to stress [8,9].

The ovary is one of the female reproductive organs. The main function is to produce a mature oocyte for fertilization and to secrete sexual hormones that facilitate the sexual development and reproductive function. Studies showed that various types of stress influenced the development of follicles and had negative effects on reproduction. For instance, chronic unpredictable mild stress induced ovarian insufficiency by interrupting lipid homeostasis. It also inhibited follicle development and increased the follicular atresia [10,11]. Chronic intermittent cold stress decreased the number of preantral healthy follicles [12]. Chronic unpredictable stress decreased the number of retrieved oocytes and impaired the developmental potential of oocytes through regulating the expression of brain-derived neurotrophic factor (BDNF) in mouse ovaries [13]. Restraint stress during pregnancy lead to reduced pregnancy rates, impaired function of the corpus luteum, and a reduced litter size in both mice and rats [14,15,16]. Chronic heat stress increased the number of atretic follicles and decreased the serum estradiol [17]. Chronic predator stress decreased the primordial follicle number. The ovarian reserve was also decreased in the chronic psychological stress model established by the scream sound [18,19]. Chronic stress also decreased the oocytes in in vivo maturation and decreased the percentage of in vitro matured oocytes, due to the irreparable damage of the cumulus cells [20].

P2X7 receptor (P2X7R) is a non-selective cation-gated channel. Adenosine triphosphate (ATP) is its ligand. P2X7R is widely distributed in various tissues and up-regulated in a variety of pathological states [21]. Activation of P2X7R leads to the assembly of NLRP inflammasome and the release of IL-1β, IL6, TNF-α and other inflammatory mediators, which participate in inflammatory response and immune response, and induce cell damage and even apoptosis. Studies have shown that the P2X7R is involved in stress-induced depression and other phenotypes [22,23,24]. Exposure of mice to chronic unpredictable mild stress enhances P2X7R expression in the hippocampus and medial prefrontal cortex. The anti-depressant effects of clemastine, ketamine, and imipramine are associated with reduced P2X7R levels in the hippocampus [25,26,27].

In this study, we established two kinds of chronic stress models, including the chronic restraint-stress and chronic cold-stress model to explore the pathological effect of chronic stress on ovarian tissues and reveal the role of P2X7R in the chronic stress-induced ovarian response.

## 2. Materials and Methods

### 2.1. Animals and Experimental Grouping

Kunming female mice were purchased from Animal Center of Jiangxi Traditional Chinese Medicine University. The mice were housed in a temperature- and light-controlled facility with free access to water and food. The animal studies and operation procedures were approved by the Ethical Committee of Nanchang University (NCULAE-202209280022).

For the chronic restraint stress model, four-week-old Kunming mice of comparable body weights were randomly divided into two groups: control group (n = 10) and chronic restraint stress group (*n* = 10). For the chronic restraint stress group, the mice were put into a 50 mL centrifuge tube for 4 h every day (from 1 p.m. to 5 p.m). The 50 mL centrifuge tube is filled with holes to ensure air circulation and the smooth breathing of mice in the centrifuge tube. The mice can move back and forth freely, but cannot turn around. For the control group, the mice were kept in cages and fasted during the corresponding period. The mice were weighed every week and recorded for 8 weeks.

For the chronic cold stress model, six-week-old mice of comparable body weights were randomly divided into two groups: control group (*n* = 10) and chronic cold stress group (*n* = 10). The mice for the cold stress group were kept in 4 °C room for 4 h each day. For the control group, the mice were kept in a 26 °C room during the corresponding period. The mice were weighed every week and recorded for 8 weeks.

### 2.2. Vaginal Smear

Vaginal secretions were collected daily for cytological monitoring at 7 a.m. in the last 2 weeks of establishing chronic restraint and the cold stress model. The swabs were pre-soaked in saline, and were gently inserted into the vagina of the mice. Then, the secretions were applied to the slides and fixed immediately in 95% ethanol. Giemsa staining solution A and solution B were added separately to stain cells for ten minutes. After rinsing with water, the slides were observed under the microscope.

### 2.3. Superovulation

In order to explore whether P2X7R signaling mediates abnormal ovulation in chronic stress groups, we conducted superovulation experiments. The mice were randomly divided into four groups (n = 6): control group, chronic restraint or cold stress group, A-438079 HCL (GC17212, GLPBIO, Montclair, CA, USA) group (P2X7R antagonist) and chronic restraint or cold stress+ A-438079 HCL group. The chronic stress model was established as previously, for 8 weeks. A total of 5 IU pregnant mare’s serum gonadotropin (PMSG) was injected intraperitoneally to induce follicular maturation, followed by 5 IU human chorionic gonadotropin (hCG) after 48 h to induce the ovulation. For the A-438079 HCL group and chronic restraint or cold stress+ A-438079 HCL group, 5 mg/kg A-438079 HCL were injected intraperitoneally for three days during superovulation. For the control group and chronic-restraint or cold-stress group, the same amount of saline was injected intraperitoneally for three days.

In addition, we also observed the effect of BzATP (GC15898, GLPBIO, CA, USA) (P2X7R agonist) and A-438079 HCL on the superovulation rate of normal healthy mice. The mice were divided into four groups (n = 6): control group, BzATP group, A-438079 HCL group and BzATP + A-438079 HCL group. The superovulation procedures were described previously.

### 2.4. Hematoxylin–Eosin (H&E) Staining

After establishing the mouse model, ovarian samples (n = 10) were fixed in 4% paraformaldehyde for 24 h and embedded in paraffin. Five-micrometer sections were cut. After deparaffinization and rehydration through degraded ethanol, the slides were stained with hematoxylin and eosin. to examine the histopathological change.

Under the microscope, we identify the corpora lutea by their structural characteristics in stained sections, where they appear as round or oval structures. We count the number of corpora lutea at their largest cross-sectional area.

### 2.5. Sirius Red Staining

In accordance with the Sirius Red Staining Kit (G1472, Solarbio, Beijing, China) instruction, the 5 μm sections (n = 10) were deparaffinized, rehydrated, and stained with sirius red staining solution for 1 h. The sections were re-immersed in alcohol and xylene, and sealed with neutral gum.

### 2.6. Granulosa Cell Culture and Treatment

The 21-day-old Kunming female mice (n = 6) were injected intraperitoneally with 5 IU PMSG. After 46 h, the ovaries were collected in basic culture medium consisting of DMEM/F12 with 5% fetal bovine serum (FBS), 100 IU/mL streptomycin and 100 IU/mL penicillin. The ovaries were punctured with a 25-gauge needle to release granulosa cells. After centrifugation for 5 min at 1200 rmp, the granulosa cells were resuspended in basic culture medium, seeded in 6-well plates and cultured in a humidified atmosphere containing 5% CO_2_ at 37 °C overnight, for adhesion. After the granulosa cells grew to 80%, the cells were treated with different reagents as indicated, including 100 µM A-438079 HCL or 100 µM BzATP, for 48 h. RNA and protein were extracted for further analysis.

### 2.7. Cumulus–Oocyte Complex (COC) Culture and Treatment

The 21 day-old Kunming female mice (n = 6) were intraperitoneally injected with 5 IU PMSG. After 46 h, the ovaries were punctured under a stereomicroscope with sterile 27-gauge needles. COCs were picked up and washed twice in preheated culture medium, by oral pipette. The COCs were cultured in a 50 µL drop covered with paraffin oil. The culture media included MEM-ALPHA, 5% FBS, 20 mM sodium pyruvate, 3 mg/mL BSA, 100 IU/mL penicillin and 100 IU/mL streptomycin. The experiment grouping included the control group, the 10 ng/mL EGF group, the 100 µM BzATP group and the 10 ng/mL EGF + 100 µM BzATP group. The cumulus expansion was observed after treatment with indicated reagents for 18 h. COCs were photographed with an inverted microscope, and the area of cumulus expansion was measured using Image J 2.15.0 software to evaluate the cumulus expansion.

### 2.8. Immunohistochemistry

After deparaffinization and rehydration through degraded ethanol, the slides received antigen retrieval in 10 mM sodium citrate buffer for 20 min. The sections were inactivated through 3% H_2_O_2_ for 10 min, incubated with 3% BSA to block nonspecific binding, and were incubated with primary P2X7R antibody (1:400 dilution, 28207-1-AP, Proteintech, Wuhan, China) overnight at 4 °C. After washing with phosphate-buffered saline (PBS), the secondary antibody was incubated for 30 min at room temperature, and the color was developed with 3,3-diaminobenzidine for 2 min. The sections were counterstained with hematoxylin for 30 s and sealed with neutral gum.

### 2.9. Corticosterone and Progesterone Determination by ELISA

Concentrations of corticosterone and progesterone in the serum were measured using an ELISA kit (E-OSEL-M0001, Elabscience Biotechnology Co., Wuhan, China; E-OSEL-M0006, Elabscience Biotechnology Co., Wuhan, China), according to the manufacturer’s instructions. Briefly, 100 μL sample or standards were added to wells, and the samples were incubated at 37 °C for 90 min. Then, the samples were incubated with HRP-conjugated antibody for 60 min at 37 °C, followed by incubation with substrate solution for 15 min at 37 °C, and the reaction was stopped by the addition of stop solution. Absorbance at 450 nm was recorded on an ELISA plate reader.

### 2.10. RNA Extraction and Real-Time PCR

The total RNA was extracted by TRIzol reagent (ET121-01, TransGen, Beijing, China), according to the manufacturer’s instructions. The purity and concentration of RNA were determined using Nanodrop. A total of 200 ng total RNA was used to synthesize the cDNA using the EasyScript One-Step gDNA Removal and cDNA Synthesis SuperMix (TransGen, Beijing, China), according to the manufacturer’s instruction. Real-time quantitative polymerase chain reactions (qPCRs) were performed using the QuantiNova SYBR Green PCR Kit (QIAGEN, Bochum, Germany). The PCR primer sequences are shown in Table 1. The gene expression levels were standardized to the levels of β-actin and evaluated using the 2−ΔΔCT method.

### 2.11. Western Blot

Total protein was extracted by RIPA lysis buffer containing protein phosphatase inhibitor and protease inhibitor. Equal amounts of total protein were separated by sodium dodecyl sulfate–polyacrylamide gel electrophoresis and transferred to the PVDF membrane. The blot was blocked with 5% skim milk for 2 h at room temperature and incubated with the corresponding primary antibodies, P2X7R (1:1000 dilution, 28207-1-AP, Proteintech, China), TGF-β1 (1:1000 dilution, 21898-1-AP, Proteintech, China), MMP2 (1:1000 dilution,10373-2-AP, Proteintech, China), and β-actin (1:5000 dilution,81115-1-RR, Proteintech, China), overnight at 4 °C. After 3 washes with TBST, the blot was incubated with HRP-labeled secondary antibodies for one hour at room temperature. The target bands were visualized using a chemiluminescent detection kit. The intensity of the bands was quantified by Bio-Rad Image Laboratory software (Version 6.0).

### 2.12. Statistical Analysis

All data were statistically analyzed using GraphPad Prism 7.00. The data were shown in the form of the mean and standard error of the mean (SEM). Statistical comparison between the two groups was performed by independent sample *t* test. One-way analysis of variance followed by the Student–Newman–Keuls test was used for statistical comparisons among multiple groups. *p* < 0.05 and *p* < 0.01 were considered statistically significant.

## 3. Results

### 3.1. The Effects of Chronic Stress on Body Weight, Serum Corticosterone Concentration and Estrous Cycle in Mice

While establishing the stress model, changes in body weight were monitored. In the chronic restraint stress group, the increase in body weight was slower than that of the control group (Figure 1A). However, there was no obvious change in body weight in chronic cold stress models (Figure 1B). The HPA axis is one of the most important stress regulation systems [8]. We detected the serum corticosterone (CORT) concentration and found that the CORT level was significantly increased in chronic restraint-stress mice (Figure 1C), while there was no significant difference in chronic cold-stress mice (Figure 1D).

To observe the effect of chronic stress on reproduction, we monitored the changes in the estrous cycle of mice. Mice in the control group had regular estrous cycles, while mice in the stress group were irregular. The late-estrus and estrus-interval stage were prolonged in the chronic restraint-stress mice and the late-estrus stage was prolonged in the chronic cold-stress mice (Table 2).

### 3.2. The Corpus Luteum Number and Progesterone Were Decreased in the Chronic Stress Model

We found that the number of corpora lutea was greatly decreased in two chronic stress models (Figure 2A,B). The number of corpora lutea indirectly reflects the ovulation. So, we inferred that chronic stress influenced the ovulation. We also detected the serum progesterone level and found that there was significant decrease in progesterone concentration in chronic restraint stress and chronic cold stress (Figure 2C).

### 3.3. The Expression of P2X7R Was Increased in Ovaries of Chronic Stress Mice and P2X7R Antagonist Partially Rescued the Ovulation Rate of Chronic Stress Mice

Studies showed that P2X7R is involved in a variety of pathological changes induced by chronic stress [22]. We examined the expression of P2X7R in two chronic stress models and found that P2X7R was obviously increased in the chronic restraint-stress and chronic cold-stress model, implying that the abnormal expression of P2X7R may contribute to the abnormal ovulation (Figure 3A–C). To further prove the effect of P2X7R signaling on abnormal ovulation in chronic stress mice, we examine the ovulation rate after treatment with P2X7R antagonist A-438079 HCL. PMSG and hCG were injected to affect the ovulation. We found that the superovulation rate was obviously decreased in the chronic-restraint and cold-stress model. Pretreatment with P2X7R antagonist A-438079 HCL partially rescued the ovulation rate of the chronic stress model (Figure 3D,E). We also observed the effect of P2X7R agonist BzATP on the superovulation rate of normal healthy mice and found that BzATP inhibited the ovulation rate. A-438079 HCL partially rescued the ovulation rate induced by BzATP (Figure 3F). These results imply that P2X7R signaling is involved in ovulation and the increase of P2X7R expression in the chronic stress model contributes to the decreased number of corpora lutea.

### 3.4. P2X7R Signaling Was Involved in Cumulus Expansion by Regulating NPPC Expression

Cumulus expansion is a key step for ovulation. In order to further explore the effect of the P2X7R signaling on ovulation, COCs were cultured and treated with P2X7R agonist BzATP. Epidermal growth factor (EGF), one main promoting factor of cumulus expansion, served as positive control. We found that EGF successfully induced cumulus expansion, while BzATP inhibited cumulus expansion induced by EGF (Figure 3A). Various factors secreted by granulosa cells are involved in the process of cumulus expansion, including Amphiregulin (AREG), Epiregulin (EREG) and Betacellulin (BTC) [28,29]. These factors act on EGF receptor (EGFR) in cumulus cells to promote cumulus expansion and oocyte maturity [30]. Natriuretic peptide precursor C (NPPC) produced in mural cells acts on NPR2 expressed in cumulus cells and plays a negative regulatory role in cumulus expansion and oocyte maturity [31]. We cultured granulosa cells and treated them with P2X7R antagonist A-438079HCL. Real-time PCR was used to examine the expression of these factors and we found that NPPC was down-regulated after treatment with A-438079HCL and there was no influence on the expression of EGF, AREG, EREG, BTC and NPR2 (Figure 4B). P2X7R agonist BzATP also up-regulated the expression of NPPC, and A-438079HCL prevented the effect of BzATP on NPPC expression (Figure 4C,D). These results suggest that NPPC is one target gene of P2X7R signaling which may contribute to the abnormal ovulation of the chronic stress model.

### 3.5. Ovarian Fibrosis Was Increased in Chronic Stress Model

The abnormal deposition of collagen and other ECM components in follicles results in abnormal follicle development and abnormal ovulation [32]. We further examined the fibrosis change in ovaries through sirius red staining. Sirius red staining showed that ovarian fibrosis was increased in ovaries of both chronic stress models (Figure 5A). We found that TGF-β1 was up-regulated and MMP2 was down-regulated in chronic-restraint and cold-stress mice (Figure 5B,C). We also examined the influence of P2X7R signaling on the expression of fibrosis regulatory factors in granulosa cells, and found that P2X7R agonist BzATP promoted the expression of TGF-β1 and decreased the expression of MMP2. P2X7R antagonist A-438079HCL rescued the effect of BzATP on TGF-β1 and MMP2 (Figure 5D). These results suggest that the increase in P2X7R may also mediate the fibrosis change in chronic-restraint and cold-stress mice.

## 4. Discussion

Clinical studies showed that the number of women with irregular menstruation, amenorrhea, and even infertility, caused by chronic psychological stress was increasing year by year [33,34]. Animal studies found that chronic stress had adverse effects on female fertility [20,35]. In this study, we established a chronic restraint-stress and chronic cold-stress mouse model and found that the estrous cycle of mice was prolonged in both the chronic stress-model mice. The number of corpora lutea was decreased significantly. The number of corpora lutea indirectly reflects the ovulation, suggesting that chronic stress influences ovulation.

Neuroendocrine factors are the main regulating factors of follicle development and ovulation. Gonadotropin-releasing hormone (GnRH) is released rhythmically by the ventromedial nucleus of the hypothalamus. It controls and regulates the synthesis and release of the follicle-stimulating hormone (FSH) and Luteinizing hormone (LH) of the pituitary gland through the pituitary portal circulation [36]. Ovaries are the target gland of the pituitary hormone. Follicular development and ovulation are strictly regulated by the hypothalamic–pituitary–ovarian axis (HPO). However, the HPO axis is not one important stress response system. The hypothalamic–pituitary–adrenocortical (HPA) axis is one of the most important stress regulation systems [8]. Studies showed that the HPA axis can influence the HPO and indirectly regulate female reproduction. The adrenocorticotropic hormone (ACTH), corticotropin-releasing hormone (CRH) and cortisol can inhibit the synthesis of GnRH in the hypothalamus under stress, and cortisol can also inhibit the secretion of LH in the pituitary gland, affecting follicle development and ovulation [37,38,39,40]. CRH also plays a regulatory role in the local ovary, promoting the activation of a large number of primordial follicles [41], and can also affect the secretion of estrogen and progesterone [39]. Chronic stress inhibits follicular development and promotes follicular atresia, and gonadotropin treatment can partially reverse this phenotype [11]. In this study, we found that the CORT level was significantly increased in chronic restraint-stress mice, suggesting that the HPA axis is activated under chronic restraint-stress mice. However, there was no significant difference in CORT level in chronic cold-stress mice. Besides the HPA axis, the autonomic nerve system is another important stress regulating system [42]. Studies have shown that sympathetic activation induced by chronic stress plays important roles in cardiovascular diseases, aging, obesity, immunity and other pathological processes [43,44]. Studies also showed that chronic cold stress can only activate the sympathetic nerve system, and they rule out the effect of the HPA axis. Considering that both the HPA axis and the sympathetic nervous system contribute to the pathological changes induced by chronic restraint stress, we infer that the abnormal ovulation in the chronic restraint stress model is associated with sympathetic activation and HPA axis activation. However, abnormal ovulation in the chronic restraint stress model is only associated with sympathetic activation.

The P2X7R is a non-selective cation-gated channel, and widely distributed in various tissues. Triphosadenine (ATP) is its ligand. Studies showed that the expression of P2X7R is up-regulated in a variety of pathological states and P2X7R activation is involved in stress-induced depression and other phenotypes [24]. ATP is one important neurotransmitter released from sympathetic nerve endings, besides norepinephrine [45,46,47]. The release of ATP upon sympathetic nerve activation can mediate various effects, such as vasoconstriction [48,49], and urinary sodium excretion by the kidney [50]. In this study, we detected the significant increase in P2X7R expression in ovaries of the two stress models, suggesting that the activation of the P2X7R pathway may be involved in the chronic stress-induced ovarian pathological process. We employed the P2X7R antagonist A-438079 HCL to prevent this signaling, to observe the effect of blocking P2X7R signaling on ovulation. We found that the ovulation rate in chronic stress-model mice was significantly lower than that of the control mice. Treatment with the P2X7 antagonist partially reversed the ovulation rate. BzATP, the P2X7R agonist, can also inhibit the superovulation rate of normal healthy mice. These results suggest that the increase in P2X7R contributes to the abnormal ovulation process in the chronic stress model.

For ovulation, the maturation of follicles and cumulus expansion are key events. In mature follicles, granulosa cells differentiate into two subsets: cumulus cells close to the oocyte and mural granulosa cells close to the follicle wall. To complete the process of ovulation, the communication between mural granulosa cells and cumulus cells is needed. Two kinds of signaling molecules play important roles in the communication between mural granulosa cells and cumulus cells. Mural granulosa cells can secrete natriuretic peptide precursor C (NPPC), while cumulus cells express natriuretic peptide receptor 2(NPR2). NPPC binds to NPR2 and upregulates the concentration of cGMP in cumulus cells, which enters into the oocytes through gap junctions, arrests oocyte meiosis, and ensures follicle maturation [31]. Three kinds of epidermal growth factors (EGFs) including amphiregulin (AREG), epiregulin (EREG) and betacellulin (BTC), produced by mural granulosa cells and induced by the LH peak, acted on EGFR expressed on cumulus cells, and induced cumulus expansion. EGF signaling also inhibits cGMP level, closes gap junctions between cumulus cells and oocytes, reduces cGMP in oocytes and restores meiosis [30,51]. Simultaneously, LH- and EGF-pathway activation during ovulation inhibited the signaling of NPPC/NPR2 [52,53]. We found that activation of the P2X7R pathway significantly inhibited EGF-induced cumulus expansion. The P2X7 agonist BzATP promoted the expression of NPPC and the P2X7R inhibitor decreased the expression of NPPC, suggesting that the P2X7R signaling pathway is involved in the ovulation process through regulating NPPC expression. Combined with the increased expression of P2X7R in the chronic stress model, we infer that the high expression of P2X7R influences the communication between mural granulosa cells and cumulus cells, and affects the cumulus expansion and ovulation (Figure 6).

In addition, studies have reported that P2X7R activation also mediates fibrosis in multiple tissues and organs, including the heart, liver, and kidney. P2X7 antagonists have become the target of anti-fibrotic therapy [54,55]. Ovarian fibrosis pathological change happened in, premature ovarian failure and other ovarian diseases. The expression of fibrosis factors TGF-β1, MMPs and TIMPs is also unbalanced in these diseases. The abnormal deposition of collagen and other ECM components in ovaries results in ovulation disorders and abnormal follicle development [32]. In this study, we found that ovarian fibrosis was significantly increased in the two chronic stress models. The expression of TGFβ1, one fibrosis-promoting factor, was up-regulated and MMP2 was decreased. P2X7R signaling activation also promoted the expression of TGF-β1 and decreased the expression of MMP2 in granulosa cells. These results suggest that abnormal expression of P2X7R may also lead to ovarian fibrosis by influencing the expression of fibrosis factors. The increase in fibrosis may also contribute to the abnormal ovulation in chronic stress mouse models.

## 5. Conclusions

In conclusion, these results demonstrated that P2X7R expression was increased in ovaries of chronic restraint-stress and chronic cold-stress mouse models. The increased P2X7R signaling influenced cumulus expansion and the ovulation process through regulating NPPC expression. The increased ovarian fibrosis induced by P2X7R may also contribute to the abnormal ovulation of the chronic stress mouse model.

## Figures and Tables

**Figure 1 biology-13-00620-f001:**
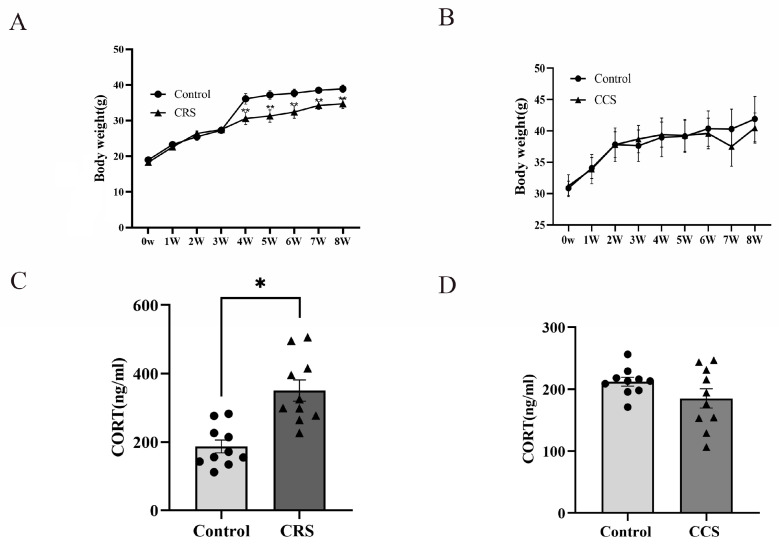
The influence of chronic stress on body weight and serum corticosterone level of mice. (**A**,**B**) show the effect of chronic restraint stress (CRS, n = 10) and chronic cold stress (CCS, n = 10) on body weight of mice. (**C**,**D**) show the effect of CRS and CCS on serum corticosterone level of mice. * indicates *p* < 0.05 and ** indicates *p* < 0.01.

**Figure 2 biology-13-00620-f002:**
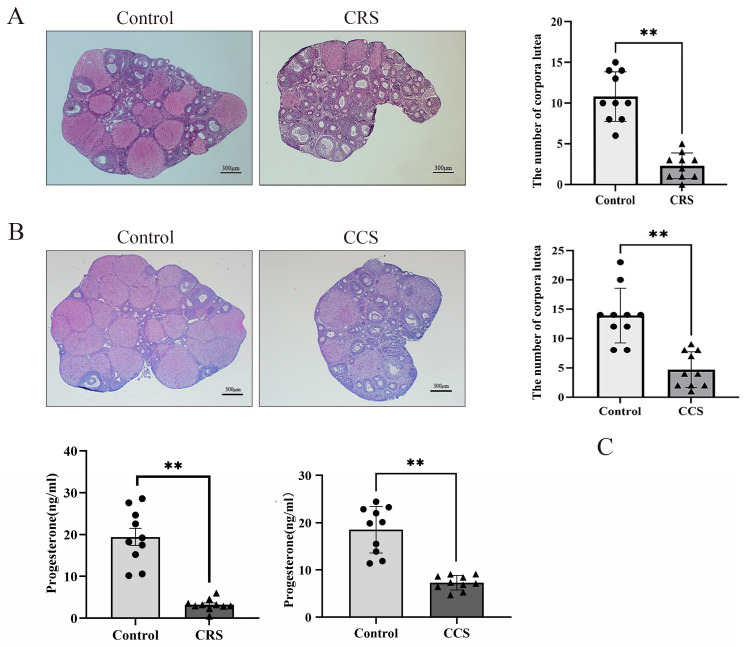
The influence of chronic stress on the number of corpora lutea and the serum progesterone of mice. After establishing the chronic stress model, HE staining was used to detect the pathological changes in ovarian tissues. (**A**) shows the effect of chronic restraint stress (CRS) on the number of corpora lutea of mice. (**B**) shows the effect of chronic cold stress (CCS) on the number of corpora lutea of mice. (**C**) shows the effect of CRS and CCS on the serum progesterone level of mice. ** indicates *p* < 0.01. Scale bar = 300 μm.

**Figure 3 biology-13-00620-f003:**
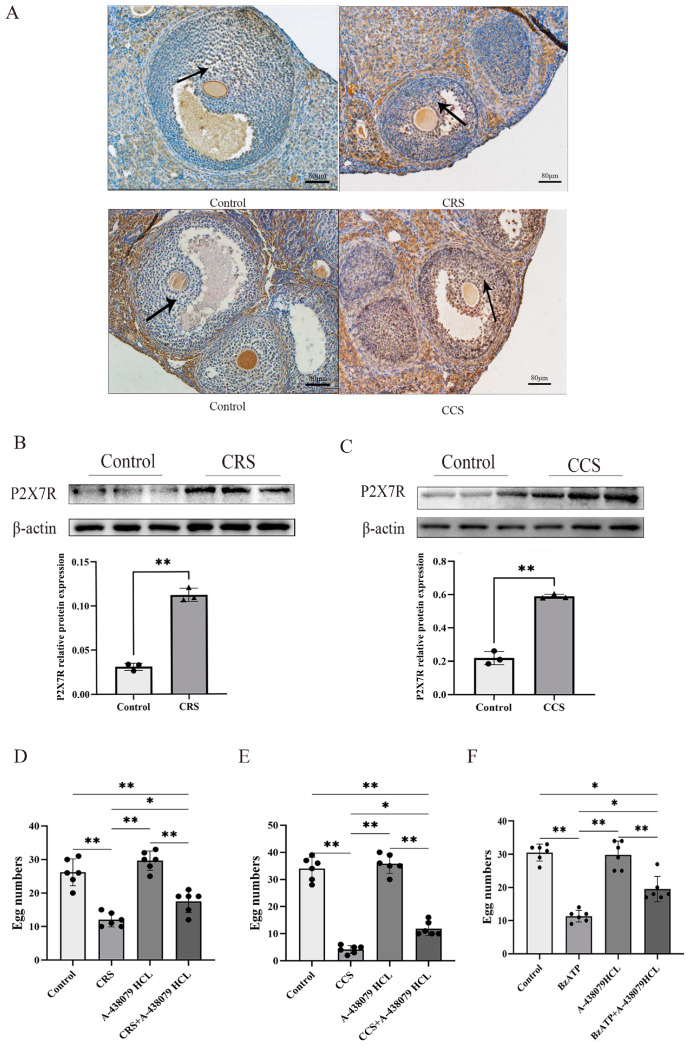
The expression of P2X7R in ovaries of chronic stress mice and the influence of P2X7R signaling on the ovulation of mice. After establishing the chronic stress model, immunohistochemistry and Western blotting were used to examine the expression change of P2X7R in ovarian tissues. (**A**–**C**) show the effect of chronic restraint stress (CRS) and chronic cold stress (CCS) on the expression of P2X7R in ovarian tissues. Superovulation was used to assess the ovulation rate of mice. (**D**,**E**) show the effect of A-438079 HCL, an antagonist of P2X7R signaling, on the superovulation rate of CRS and CCS mice. (**F**) shows the effect of BzATP and A-438079 HCL on the superovulation rate of normal healthy mice. * indicates *p* < 0.05 and ** indicates *p* < 0.01. Scale bar = 80 μm. The uncropped western blot figures were presented in Appendix A.

**Figure 4 biology-13-00620-f004:**
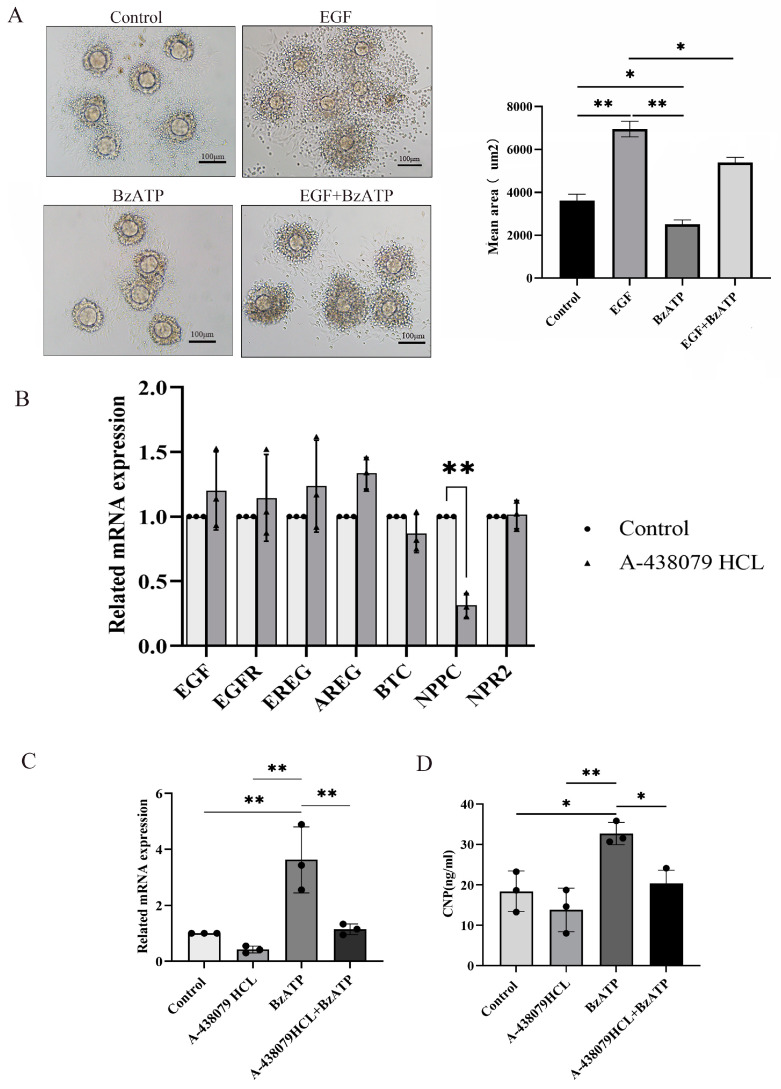
The influence of P2X7R signaling on cumulus expansion in vitro and the expression of ovulation-related factors. (**A**) shows the effect of BzATP, an agonist of P2X7R signaling, on cumulus expansion induced by EGF. EGF served as positive control of cumulus expansion. Real-time PCR was used to detect the change in expression of ovulation-related factors. (**B**) shows the effect of A-438079 HCL, an antagonist of P2X7R signaling, on the expression of ovulation-related factors in granulosa cells. We found that NPPC was down-regulated. (**C**) shows the effect of BzATP and A-438079 HCL on NPPC mRNA expression. (**D**) shows the effect of BzATP and A-438079 HCL on CNP protein level. * indicates *p* < 0.05 and ** indicates *p* < 0.01. Scale bar = 100 μm.

**Figure 5 biology-13-00620-f005:**
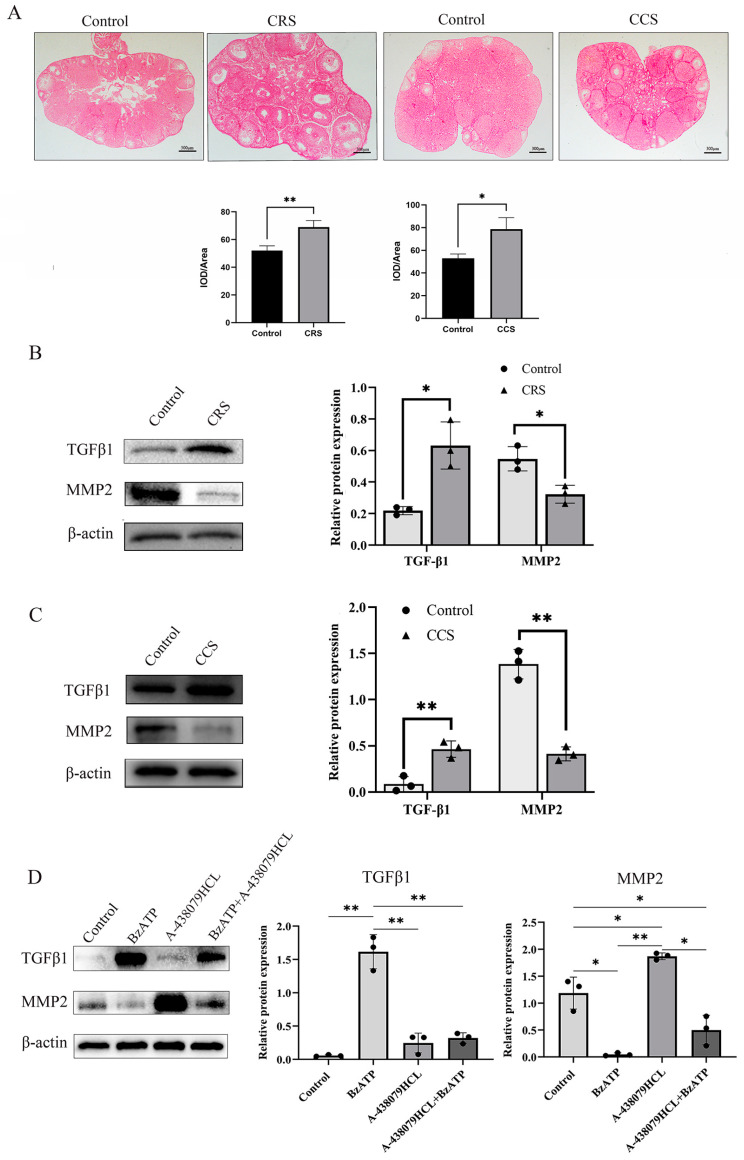
The influence of chronic stress on ovarian fibrosis. The changes in tissue fibrosis were detected by sirius red staining. (**A**) shows the representative sirius red staining of ovary in the control, chronic restraint stress (CRS) and chronic cold stress (CCS). (**B**) shows the expression of fibrosis-related factors, including TGF-β1 and MMP2 in chronic restraint stress (CRS) mice. (**C**) shows the expression of TGF-β1 and MMP2 in chronic cold stress (CCS) mice. (**D**) shows the effect of BzATP and A-438079 HCL on TGF-β1 and MMP2 expression in granulosa cells. * indicates *p* < 0.05 and ** indicates *p* < 0.01. Scale bar = 300 μm. The uncropped western blot figures were presented in Appendix A.

**Figure 6 biology-13-00620-f006:**
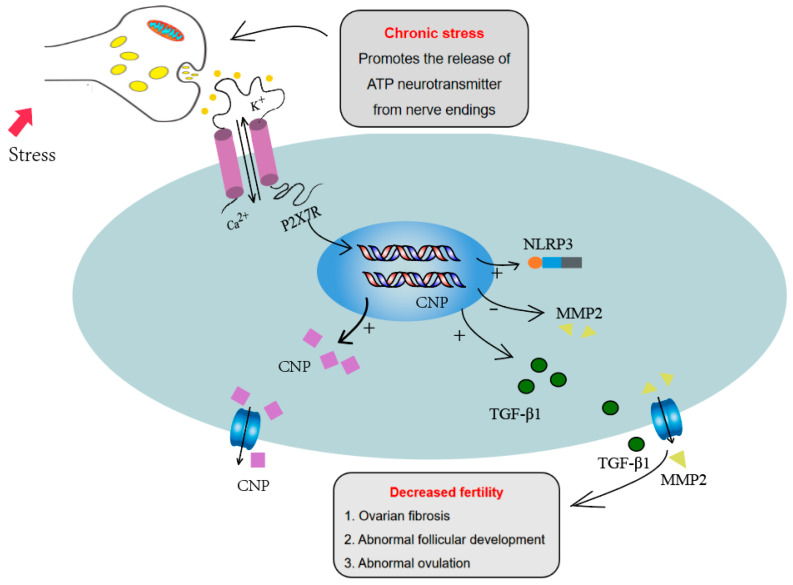
P2X7R signaling influenced the ovulation process through regulating NPPC expression and increased the ovarian fibrosis through regulating fibrosis-related factors, including IL-1β, TGF-β1 and MMP2, which may contribute to the abnormal ovulation of the chronic stress mouse model.

**Table 1 biology-13-00620-t001:** Oligonucleotides used for Real-Time PCR.

Gene	Sense and Antisense Primers
EGF	5′-ACTCCCACCTACCCTCCTA-3′ 5′-CTGAACTGGCTCTGTCTGC-3′
EGFR	5′-ACCTCTCCCGGTCAGAGATG-3′ 5′-CTTGTGCCTTGGCAGACTTTC-3′
EREG	5′-TGCTTTGTCTAGGTTCCCACC-3′ 5′-GGCGGTACAGTTATCCTCGG-3′
AREG	5′-GCTGAGGACAATGCAGGGTAA-3′ 5′-GTGACAACTGGGCATCTGGA-3′
BTC	5′-CCTCACAGCACAGTTGATGG-3′ 5′-GGTGTTCTGGTTGTGTTCCC-3′
NPPC	5′-GGTCTGGGATGTTAGTGCAGCTA-3′ 5′-TAAAAGCCACATTGCGTTGGA-3′
NPR2	5′-GCTGACCCGGCAAGTTCTGT-3′ 5′-ACAATACTCGGTGACAATGCAGAT-3′
β-actin	5′-CACGATGGAGGGGCCGGACTCATC-3′ 5′-TAAAGACCTCTATGCCAACACAGT-3′

**Table 2 biology-13-00620-t002:** Effect of chronic stress on estrous cycle in mice.

Group	n	Proestrus (d)	Estrus (d)	Late Estrus (d)	Estrus Interval (d)	Estrous Cycle (d)
Control	10	1.05 ± 0.05	1.27 ± 0.09	1.92 ± 0.15	1.21 ± 0.08	4.54 ± 0.51
CRS	10	0.81 ± 0.04	1.21 ± 0.09	2.58 ± 0.15 *	2.01 ± 0.09 *	5.81 ± 0.79 *
Control	10	1.09 ± 0.16	1.72 ± 0.23	1.36 ± 0.20	1.27 ± 0.19	5.36 ± 0.52
CCS	10	1.15 ± 0.10	1.38 ± 0.14	2.46 ± 0.29 **	1.76 ± 0.12	6.53 ± 0.63 *

CRS: Chronic restraint stress; CCS: Chronic cold stress. * means *p* < 0.05; ** means *p* < 0.01.

## Data Availability

The datasets generated and/or analyzed during the current study are available from the corresponding author on reasonable request.

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
