# Peer review of "Activation of P2X7 Receptor Mediates the Abnormal Ovulation Induced by Chronic Restraint Stress and Chronic Cold Stress"

_biology, 2024, doi:10.3390/biology13080620_

Round 1
Reviewer 1 Report
Comments and Suggestions for Authors
The submitted manuscript examined the effects of two chronic stressors, physical and cold, on reproduction impairment in female mice. The researchers used those stressors to model the impact of chronic stress on the reproductive system in human females. Because the P2X7 receptor is involved in other stress situations, the investigators reasoned it would be a logical endpoint for study in these stress models. Both stressors reduced ovulation rate, increased expression of P2X7R in ovarian tissues, inhibited cumulus cell expansion, and increased ovarian fibrosis. The effect of the stressors was partially overcome with a P2X2R inhibitor supporting their hypothesis.
The authors need to be more explicit in the Materials and Methods. Were the same mice used throughout, and did those provide ovaries for the other procedures? In the Superovulation section, the design is a two X two factorial with six mice in each group for 24 mice for one stressor. Were another 24 mice used for the second stressor, or was this repeated on the same animals? How many ovaries were used in H&E and Sirius Red staining, and when were ovaries collected and processed? How many mice were utilized for granulosa cell culture, and are the same mice used for COCs? The COCs were two x two designs. Where did the slides for immunochemistry come from? When, how, and how many samples were taken for ELISA tests? What is the source for RNA and Westerns?
The statistical analysis needs to be more precise. The analysis for some was a t-test, but the other should have been two-way, not one-way.
Comments on the Quality of English Language
L 106 change from rinse to rinsing.
L 120 change from observe to observed.
L 138 change from centrifuged to centrifuging
L 141 change to “After granulosa cells grew”
L 194 change were to are
L 205-207 Delete "Hypothalamic….and”
L 209 incomplete
L 211 Delete “Estrous cycle showed that”
L 225-226 Delete sentence
L 245-251 Delete” Various… [31].”
L 260-262 Delete sentences
L 265-267 Delete sentences
Reviewer 2 Report
Comments and Suggestions for Authors
In this study, Fan et al. investigated the role of P2X7R in ovulation under chronic stress using both established chronic restraint stress and chronic cold stress mice models. They found that P2X7R expression in the ovary remarkably increased in both chronic stress models. Additionally, activation of P2X7R signaling inhibited the cumulus expansion and promoted the expression of NPPC in granulosa cells.
Overall, the authors provided multiple pieces of evidence demonstrating the negative impact of chronic stress on ovarian function. However, one important concern is that pretreatment with a P2X7R antagonist only mildly rescued the ovulation rate, indicating that P2X7R does not significantly contribute to the ovulation defect phenotype.
Below are some specific concerns:
1. According to Materials and Methods section, four-week-old mice were used for both chronic restraint stress model and chronic cold stress model. However, the mouse body weight at 1 week varies significantly between the restraint stress model and the chronic cold stress model (~23 g vs. ~34 g, Fig. 1A and B). As the initial body weight is so high, probably that’s the reason why the authors did not observe obvious change of body weight in chronic cold stress models. Additionally, what’s the original body weight before the experiment?
2. Line 209: remove “and”.
3. How did the authors perform the corpus luteum counting? The related method should be described in the Methods section.
4. Based on the IHC images shown in Fig. 3A, P2X7R broadly expresses in various ovarian cells, including oocytes, granulosa cells, theca cells, and interstitial stroma cells. Did the authors perform a negative staining control? Which cells can express P2X7R in the ovary? Additionally, why is the P2X7R signal remarkably higher in the CCS control group compared to the CRS control group?
5. Lines 244-245: the authors should calculate the COC expansion rate before drawing their conclusion. Additionally, the MII oocyte rate should be counted.
6. Lines 264-265: quantification of fibrosis signals should be performed to support the authors’ conclusion.
7. The quantification of TGFβ1 appears inaccurate. Based on the WB image, the TGFβ1 expression level is remarkably higher in the A-438079HCL group than in the BzATP+ A-438079HCL. However, this difference is not reflected in the quantification results.
Reviewer 3 Report
Comments and Suggestions for Authors
This manuscript describes a well designed set of experiments that provide evidence for the effect of stress in increasing the synthesis and release of NPPC which activates P2X7R and causes disruption of the normal ovulation in female mice.
Apart from a few details listed below, the manuscript is generally well presented with informative background information and a clear logic for the experiments carried out. The data obtained is a useful contribution to knowledge in this field and has the potential to lead to useful medical applications.
The main point that the authors should clarify refers to line 352-356 where they try to explain the fact that restrain stress induces corticosterone release but cold stress does not, while both stress systems produce the same outcome in most other assays carried out. The authors need to clarify why the results could not be explained by activation of only the sympathetic system (and not HPA) and they should discuss what evidence they have of sympathetic activation in their study.
The rest of the discussion relates well to the results. However, it would be very useful if the authors produced a diagram where they can indicate the contribution of their work to the identification of the putative mechanisms that lead to disrupted ovulation.
Minor corrections
Line 110 change to “ two sets of four groups” (four groups for each stress model).
Line 138 change to “After centrifugation...”.
Line 189 change to “After 3 washes...”.
Line 237 change “induced” to “affected” or “reduced”.
Line 278 It is unusual to use the past tense in figure legend. “A and B showed” can be changed to “A and B show” or showed can be removed completely and use colon “A, B: The effect of ....” Whatever style the authors chose it will need to be consistent in all the figure legends.
Line 279: Figure 1 It would be better for the Y axis of A and B, and C and D to have the same scale as they present the same measurements.
Line 286 Figure 2C there is a problem with the units of the Y axis for progesterone. One is ten times higher than the other.
Line 289 Figure 3 “Related protein expression” should be “Relative protein expression”. It would actually be useful to put the name of the protein "P2X7R (relative protein expression)" This needs be corrected in several other figures
Line 304 “CNP protein level” is this supposed to be “NPPC protein level” ? what method was used to measure it?
Line 308. The staining in A is quite diffuse. The authors may want to consider adding some arrows to help the reader interpret the differences in staining . Maybe a higher magnification of a selected area of the slide would be useful
Line 337 “regulated female reproduction” (remove “the”)
Comments on the Quality of English Language
Some minor corrections are required and I have listed them in my comments above.
Round 2
Reviewer 2 Report
Comments and Suggestions for Authors
The authors have adequately addressed all the revisions requested in the previous round of review. The manuscript has been improved. I have no further comments or suggestions at this time.